# Clinical Perspectives in Addressing Unsolved Issues in (Neo)Adjuvant Therapy for Primary Breast Cancer

**DOI:** 10.3390/cancers13040926

**Published:** 2021-02-23

**Authors:** Ryungsa Kim, Takanori Kin

**Affiliations:** 1Breast Surgery, Hiroshima Mark Clinic, 1-4-3F, 2-Chome, Ohte-machi, Naka-ku, Hiroshima 730-0051, Japan; 2Department of Breast Surgery, Hiroshima City Hospital, 7-33, Moto-machi, Naka-ku, Hiroshima 730-8518, Japan; ymj5014266@gmail.com

**Keywords:** breast cancer, adjuvant therapy, neoadjuvant therapy, residual tumor cells, antitumor immunity

## Abstract

**Simple Summary:**

The development of adjuvant and neoadjuvant therapies, and breast cancer surgery for primary breast cancer has led to the dramatic improvements in the survival rates of breast cancer patients over the past 50 years. However, recurrence with distant metastasis during the 10 years after surgical treatment is still seen, although not often. Current clinical perspectives are summarized to address unsolved issues in (neo)adjuvant therapy for primary breast cancer. It is necessary to elucidate the gain of antitumor immunity via anticancer agents, the enhancement of drug sensitivity by overcoming drug resistance, and the targeting of therapy based on genomic profiles, which will lead to the complete curing of primary breast cancer.

**Abstract:**

The treatment of primary breast cancer has evolved over the past 50 years based on the concept that breast cancer is a systemic disease, with the escalation of adjuvant and neoadjuvant therapies and de-escalation of breast cancer surgery. Despite the development of these therapies, recurrence with distant metastasis during the 10 years after surgical treatment is observed, albeit infrequently. Recent advances in genomic analysis based on circulating tumor cells and circulating tumor DNA have enabled the development of targeted therapies based on genetic mutations in residual tumor cells. A paradigm shift involving the application of neoadjuvant chemotherapy (NAC) has enabled the prediction of treatment response and long-term prognoses; additional adjuvant chemotherapy targeting remaining tumor cells after NAC improves survival. The activation of antitumor immunity by anticancer agents may be involved in the eradication of residual tumor cells. Elucidation of the manner in which antitumor immunity is induced by anticancer agents and unknown factors, and the overcoming of drug resistance via the targeted eradication of residual tumor cells based on genomic profiles, will inevitably lead to the achievement of 0% distant recurrence and a complete cure for primary breast cancer.

## 1. Introduction

Despite remarkable advances in adjuvant and neoadjuvant treatments for patients with breast cancer over the past 50 years, recurrence with distant metastasis remains a persistent problem; however, recurrence is rare at 10 years after surgical treatment. Adjuvant chemotherapy and endocrine therapy (ET) have reduced breast cancer recurrence, and breast cancer mortality rates have decreased over time [1,2,3]. However, the cure efficacy of adjuvant therapy is assessed based only on the occurrence of distant metastasis during the follow-up period. Several prospective randomized controlled trials (RCTs) have been conducted to develop new adjuvant therapy regimens and drugs, leading to gradual improvement of the survival rate of patients with breast cancer [4,5]. In particular, the introduction of intrinsic tumor subtype [6] and multigene assays, such as Oncotype Dx (Genomic Health Inc., Redwood City, CA, USA) [7] and MammaPrint (Agendia Inc., Amsterdam, The Netherlands) [8], has enabled the individualization of adjuvant chemotherapy. Adjuvant therapies can be classified according to identified tumor subtypes; for example, escalation therapy specifically targets human epidermal growth factor receptor 2 (HER-2) in patients with HER-2-positive breast cancer [9]. Multigene assays enable the avoidance of unnecessary chemotherapy that does not prolong survival when combined with ET (compared with ET alone) in patients with hormone receptor (HR)-positive/HER-2-negative breast cancer [10,11]. Thus, adjuvant chemotherapy can now target specific growth factors involved in survival and tumor growth in patients with high survival rates. Nonetheless, the achievement of a 0% distant metastasis rate remains far from the establishment of a cure for primary breast cancer.

The paradigm shift brought about by neoadjuvant chemotherapy (NAC) has enabled the prediction of treatment response and prognosis before surgery, depending on the tumor subtype [12]. The achievement of a pathological complete response (pCR) after NAC is associated with a better prognosis than is non-pCR in patients with HER-2-positive and triple-negative (TN) breast cancers, but not in those with the luminal subtype [13]. However, distant metastatic recurrence is seen during the follow-up period in patients with HER-2-positive and TN breast cancers, despite the achievement of pCR by NAC. The achievement of pCR may be responsible for distant recurrence due to the presence of residual tumor cells at the molecular level, as detected by digital polymerase chain reaction (dPCR) [14] and targeted dPCR [15]. This review seeks to provide a clinical perspective on some of the barriers preventing the establishment of a cure for primary breast cancer, potential strategies for the achievement of 0% distant metastasis, and methods to achieve a cure with primary treatment, including surgery, radiotherapy, and neoadjuvant and adjuvant therapies.

## 2. The Goal of Primary Breast Cancer Treatment

In general, the goal of primary breast cancer treatment is the achievement of a cure without recurrence, especially distant metastasis, and that primary treatment provides the only opportunity to cure primary breast cancer after diagnosis. Such a cure requires the eradication of all breast cancer cells by surgery, radiation, and/or adjuvant therapies, including chemotherapy and ET. Importantly, the minimal presence of residual tumor cells circulating in the blood and other organs after the surgical resection of primary tumors and axillary lymph nodes in patients with breast cancer can cause distant metastasis if these cells are not eradicated completely by adjuvant therapy. However, we have no means of evaluating the efficacy of adjuvant therapy for residual tumor cells; we have no choice but to follow patients for the detection of local or distant recurrence for 10 years after surgical treatment. Distant recurrence can occur years after adjuvant chemotherapy and during ET. Furthermore, the reasons for the failure of adjuvant therapy in some patients remain unknown. Patients with HER-2-positive and TN breast cancers in whom pCR is achieved by NAC have higher survival rates than do those in whom pCR is not achieved. The treatment of such patients with non-pCR after NAC improved survival compared with conventional treatment [16,17], suggesting that residual tumor cells can be partially eradicated with additional adjuvant chemotherapy to improve survival.

## 3. Detection of Residual Breast Tumor Cells and Targeted Therapy

Circulating tumor cells (CTCs) and circulating tumor DNA (ctDNA) in patients with metastatic breast cancer can be detected by the CellSearch System (Veridex LLC, Raritan, NJ, USA) [18], which recognizes epithelial-cell adhesion molecules (cytokeratins 8, 18, and 19), and PCR [19], respectively. The number of CTCs and percentage of ctDNA correlate with progression-free survival after treatment. One study demonstrated that a concentration of fewer than five (vs. more than five) tumor cells per 7.5 mL whole blood conferred a better prognosis after the treatment of metastatic breast cancer [18]. Another study showed that the percentage of ctDNA predicts treatment outcomes in patients with HER-2-positive metastatic breast cancer [19]. ctDNA can be detected with high sensitivity by targeted dPCR, even in patients with pCR after NAC [15]. Whether ctDNA causes recurrence with distant metastasis in patients with pCR remains unclear, but the detection of ctDNA after NAC or adjuvant chemotherapy has been associated with a high risk of recurrence in patients with early-stage breast cancer [20]. In addition, detection of minimal residual disease by cell-free DNA was strongly associated with distant recurrence in patients with stages 0–III breast cancer who received curative treatment after surgery [21]. Genomic profile analysis revealed various genomic mutations in primary breast tumors, including those of PIK3CA, AKT1, estrogen receptor 1 (ESR1), and checkpoint kinase 2, and more frequent mutations in metastatic lesions [22]. In a recent phase-II trial, tumor genomic profiles were evaluated using ctDNA for the administration of mutation-directed therapies; the mutations considered involved ESR1, HER-2, AKT1, P53, and phosphatase and tensin homolog deleted from chromosome 10 [23]. That study revealed that patients treated for advanced and recurrent breast cancers responded well to HER-2- and AKT1-directed therapy [23], suggesting that the mutation-directed targeted treatment of residual tumor cells improves survival in patients with early-stage breast cancer after NAC and adjuvant chemotherapy or during ET. The eradication of residual tumor cells and targeting of those mutations has the potential to cure primary breast cancer.

The targeting of residual tumor cells after surgical treatment may be difficult because these cells have not been characterized. However, genomic analysis of CTCs in primary tumors and blood may enable such targeting. Current RCTs involve attempts to design integrated molecular-targeting agents for adjuvant therapy, such as HER-2 targeting for HER-2-positive breast cancer [9,17]. HER-2-positive breast cancer cells that are sensitive to HER-2-targeting agents can be eradicated by treatment; for TN and luminal breast cancers, however, the residual tumor cell targets other than AKT1 are not known. For patients with ET-resistant breast cancer cells, the combined use of ET and cyclin-dependent kinase 4/6 (CDK4/6) inhibitors with adjuvant therapy may improve survival. Recent RCTs examining the use of CDK4/6 inhibitors in combination with ET in patients with HR-positive/HER-2-negative breast cancer have yielded conflicting results: palbociclib combined with standard ET did not improve invasive disease-free survival (IDFS) relative to ET alone [24], whereas abemaciclib in combination with standard ET improved IDFS relative to ET alone in patients with node-positive/HR-positive/HER-2-negative breast cancer [25]. These findings suggest that targeted therapy with CDK4/6 inhibitors is not a unique candidate for improvement of the survival of patients with early-stage breast cancer after surgical treatment (Figure 1).

## 4. Immunosurveillance and Immunoediting for Residual Breast Tumor Cells

Cancer immunosurveillance and immunoediting play important roles in cancer progression. They consist of three processes known as the three E’s: elimination, equilibrium, and escape [26]. In general, cancer cells can achieve nonautonomous proliferation by increasing genomic mutations that evade immunosurveillance by immune cells via the immunosuppressive network of the tumor microenvironment. This network is made up of myeloid-derived suppressor cells (MDSCs), such as immature dendritic cells, and of tumor-derived soluble factors, such as vascular endothelial growth factor, which induces the expression of regulatory T cells and immunosuppressive cytokines, such as interleukin 10 (IL-10). This process results in the suppression of cell-mediated immunity in tumor cells by natural killer (NK) cells and cytotoxic T lymphocytes (CTLs) [27]. The immunosuppressive network is also mediated through the immune checkpoint (IC) molecule programmed cell death ligand 1 (PD-L1), which is expressed in tumor cells, and the immune response by CTLs is down-regulated via programmed cell death 1 (PD-1) by PD-L1 in tumor cells or cytotoxic T lymphocyte–associated antigen 4 in antigen-presenting cells [28]. We do not know how breast tumor cells retained in the blood and other parts of the body survive or whether these cells are sensitive or resistant to immune cells. These residual tumor cells can be eliminated by immunoediting the host defense immunity; however, the mechanism by which they escape immunosurveillance has not been determined. One possibility is that residual tumor cells persist in a less-immunogenic, dormant, or senescent form after chemotherapy or during ET to escape attack by immune cells. The immunogenicity characteristics of residual tumor cells, however, remain unknown.

## 5. Induction of Antitumor Immunity after Neoadjuvant and Adjuvant Treatments

Treatment with anticancer agents induces crude cell death, including apoptosis, necrosis, and other types of cell death, resulting in the release of tumor antigens (TAs) and tumor-associated antigens (TAAs) that are trapped by dendritic cells and activate CTLs [29]. Immunogenic cell death (ICD), which occurs via endoplasmic reticulum stress and anticancer agent-induced apoptosis, is a key pathway in this process [30]. ICD activates a signaling pathway that breaks immune tolerance and activates TAs and TAA-specific T cells [31]. Based on this hypothesis, NAC can activate antitumor immunity and may improve survival relative to adjuvant chemotherapy. Unfortunately, in two prospective RCTs (National Surgical Adjuvant Breast and Bowel Projects B-18 [32] and B-27 [33]), the survival rate of patients who received preoperative chemotherapy was no better than that of patients who received postoperative chemotherapy. However, a subset analysis in the B-18 trial revealed better disease-free survival (DFS) of patients with HR-negative cancer aged < 50 years who received preoperative doxorubicin and cyclophosphamide (AC) than of those who received postoperative AC [32]. In the B-27 trial, preoperative AC followed by docetaxel yielded a higher doublet pCR rate, a tendency toward improved DFS, and significantly better relapse-free survival (RFS) compared with AC alone [34]. The improved DFS and RFS in patients treated preoperatively with anthracyclines and taxanes may be due to increased pCR rates relative to those in patients receiving anthracyclines alone. Nevertheless, preoperative chemotherapy may induce a specific immune response that activates antitumor immunity involving the shrinkage of the primary tumor and eradication of metastatic tumor cells in the axillary lymph nodes.

Tumor-infiltrating lymphocytes (TILs) play an important role in the achievement of a therapeutic response after NAC in patients with TN and HER-2-positive breast cancer. High percentages of TILs before NAC were associated with a high pCR rate and favorable long-term outcomes in such patients [13]. In contrast, patients with luminal disease in whom pCR was achieved did not necessarily have better prognoses than those in whom pCR was not achieved due to the effects of ET [13]. Thus, the presence of TILs before NAC plays an important role in the induction of an immune response after treatment with anticancer agents. Recent studies have shown that increased peripheral natural killer (pNK) cell activity in the blood and increased NK levels in primary breast tumors in the presence of TILs prior to NAC are important for the induction of therapeutic effects on primary lesions and metastatic lymph nodes [35,36]. These results suggest that the induction of local and systemic immune responses and their collaboration are important for the achievement of therapeutic effects after NAC.

Another approach to the enhancement of the immune response is to down-regulate the immunosuppressive network by combining IC inhibitors, such as anti-PD-1 and anti-PD-L1 antibodies, with anticancer agents. The combined administration of IC inhibitors with anthracyclines and taxanes increased pCR rates in patients with TN and luminal breast cancer in the neoadjuvant setting [37,38]. However, whether these PD-1 and PD-L1 inhibitors are effective for HR-positive/HER-2-negative breast cancer in the adjuvant setting is unknown. Despite the efficacy of IC inhibitors used with anthracyclines and taxanes in the neoadjuvant setting, atezolizumab combined with atezolizumab and carboplatin/nab-paclitaxel chemotherapy did not increase the pCR rate in patients with high-risk early-stage TN breast cancer [39]. Furthermore, the addition of durvalumab to nab-paclitaxel followed by epirubicin/cyclophosphamide administration as neoadjuvant treatment did not have the same degree of benefit [40]. Similar conflicting findings have been reported for the combination of atezolizumab with different agents, such as nab-paclitaxel and paclitaxel, for the treatment of metastatic disease [41]. As in the metastatic setting, many RCTs are currently being conducted to evaluate the therapeutic efficacy of IC inhibitors used in combination with anticancer agents in the neoadjuvant and partially adjuvant settings [42,43]. The effects of different combinations of anticancer agents and IC inhibitors remain confusing due to discrepancies among trial results. The therapeutic efficacy of IC inhibitors in the neoadjuvant setting appears to require >1% PD-1 expression in immune cells, and these inhibitors may induce PD-L1 expression. Thus, we must await further data from ongoing clinical trials to determine which IC inhibitors are effective and suitable for use in combination with anticancer agents. Importantly, further studies are required to investigate the activation of immune responses by IC inhibitors in patients who have responded to them.

In adjuvant chemotherapy, the main effect of anticancer agents on residual tumor cells is cytotoxic or cytostatic, resulting in cell death; this effect depends on the class of chemotherapy, tumor subtype, and sensitivity to tumor cells. In this context, whether antitumor immunity can be evoked after residual tumor cell death induced by anticancer agents is unclear. However, anticancer agents such as paclitaxel and docetaxel stimulate an immune response that increases immunostimulatory cytokines, such as interferon γ, IL-2, and IL-6, and pNK cell activity, and inhibits the functions of MDSCs and regulatory T cells (Tregs) in patients with breast cancer [44,45]. Other anticancer agents, such as AC, also stimulate immunity through ICD, which activates T cells and inhibits MDSCs and Tregs [46]. Thus, with the exception of cytotoxic and cytostatic effects, anticancer agents themselves stimulate cell-mediated immunity, which contributes to the potentiation of their antitumor effects to eradicate residual tumor cells after the surgical treatment of breast cancer. However, whether such a stimulated immune response leads to the eradication of residual tumor cells, leading to the curing of breast cancer, or whether evasion of the immune response is involved in resistance to tumor cells that cause recurrence with distant metastasis, remains unclear.

In adjuvant ET, treatment with antiestrogens, selective estrogen modifiers, or aromatase inhibitors induces apoptosis or senescence as an irreversible cessation of breast cancer cell growth [47]. In the case of apoptosis, certain T cells may be stimulated via ICD by presenting TAs and TAAs to eradicate residual tumor cells, whereas senescence as a dormant state confers resistance to ET due to the presence of antiapoptotic proteins, such as BCL-2 and BCL-XL [48]. This accumulation of senescent cells attracts immune cells that cause inflammation, and the reawakening of resistant tumor cells leads to distant recurrence. Combination therapy with venetoclax, a selective inhibitor of BCL-2, and a CDK4/6 inhibitor may help to overcome resistance to ET and reduce the incidence of distant metastasis [49].

## 6. A Primary Breast Cancer Cure after (Neo)Adjuvant Chemotherapy

The ability to cure breast cancer seems to depend on two things: the susceptibility of tumor cells to anticancer drugs and the induction and gain of antitumor immunity via anticancer drugs to prevent distant recurrence. These factors can be addressed to some extent by NAC, but several factors related to drug-resistant cells may prevent the induction and gain of antitumor immunity by anticancer drugs after NAC. This situation may be the same for adjuvant chemotherapy. The factors that regulate antitumor immunity to eradicate residual tumor cells after cancer treatment remain unidentified (Figure 2). Nevertheless, the response to anticancer agents correlates with tumor shrinkage, cell death, and the induction of antitumor immunity. Furthermore, the activation of local and systemic immune responses in the presence of TILs prior to NAC is involved in the improvement of the therapeutic effect. The question of how the induction of antitumor immunity with anticancer agents leads to the complete curing of primary breast cancer remains. Even when a clinically complete response is achieved, a pCR and the eradication of residual tumor cells are not guaranteed. Unknown factors may be involved in the critical point leading to a cure in conjunction with the induction and gain of antitumor immunity to prevent distant recurrence. The hypothesized relationships of tumor volume to cell death and antitumor immunity after (neo)adjuvant chemotherapy for primary breast cancer are shown in Figure 3.

## 7. Tumor Heterogeneity and Drug Resistance

Tumor heterogeneity and resistance to anticancer agents and ET are accepted as important obstacles to the curative treatment of primary breast cancer. Two models of tumor heterogeneity have been proposed [50]: a clonal evolution model, in which random genetic mutations and clonal selection cause cellular heterogeneity in breast tumors [51]; and a stem cell model involving the diversity and hierarchical organization of tumor cells generated by breast cancer stem cells (BCSCs) [52]. In both models, the tumor microenvironment plays an important role in the evolution of breast cancer cells. BCSCs form a heterogeneous population of breast cancer cells of various types with the ability to self-renew and differentiate, and are involved in the cellular origin, tumor maintenance, and development of breast cancer [53]. Importantly, BCSCs are clinically resistant to anticancer agents and molecular targeted agents and are associated with cancer recurrence and poor prognosis [54,55]. Two distinct subpopulations of BCSCs are commonly detected by molecular markers (CD44 positivity and CD24 negativity) and low cell counts (CD44^+^/CD24^−/low^) [56], as well as aldehyde dehydrogenase 1 (ALDH1) positivity (ALDH1^+^) [57]. Various other BCSC markers have been identified, suggesting a different hierarchical organization of breast cancer and the dynamic nature of BCSC regulation by the tumor microenvironment. CD44^+^/CD24^−/low^ cells are defined as BCSCs in epithelial–mesenchymal transition (EMT), and ALDH1^+^ cells are defined as BCSCs in mesenchymal–epithelial transition (MET) [58]. Transient EMT–MET switching has been observed in metastatic tumor cells [59], and these two subgroups may represent two dynamic states of BCSCs. In considering targeted therapies, key signaling pathways in BCSCs, such as Wnt [60], Notch [61], and Hedgehog [62], may be important targets to overcome drug resistance. Inhibitors that block these signaling pathways have been developed to target BCSCs and are currently being studied in clinical trials for the treatment of breast cancer. Another strategy to inhibit the growth and function of BCSCs involves microRNAs [63]. Preclinical studies have shown that several microRNAs suppress the clonogenicity of BCSCs and inhibit tumor growth and metastasis through signaling pathways [64,65,66]. Further studies are needed for the development of a microRNA-based strategy for the treatment of breast cancer.

BCSCs differ among breast cancer subtypes and are much more likely to be present in TN and HER-2 breast cancers than in the luminal type. TN breast cancer showed BCSC enrichment associated with poor prognosis due to the development of drug resistance [67]. Poly-ADP-polymerase (PARP) inhibitors have favorable therapeutic effects on BRCA1-mutated breast cancer, and preclinical studies have shown that the PARP inhibitor olaparib significantly reduced the proportion of BCSCs, suggesting that it acts against these cells [68]. In BRCA1-mutated TN breast cancer, however, some BCSCs showed relative resistance to olaparib treatment [69]. HER-2-positive breast cancer is characterized by a high proportion of epithelial ALDH1^+^ BCSCs [52]. HER-2 is an important regulator of BCSCs, and HER-2 inhibitors such as trastuzumab, pertuzumab, lapatinib, and trastuzumab emtansine have shown some clinical efficacy. However, a proportion of HER-2-amplified breast cancers continue to develop drug resistance, eventually due to phosphatase and tensin homolog loss and the activation of PIK3CA mutations [70]. Further studies are needed to elucidate the precise manner in which BCSCs develop resistance to HER-2 inhibitors. Among luminal breast cancers, type A has the smallest proportion of BCSCs, associated with a good prognosis, whereas type B has a larger proportion of BCSCs and a poorer prognosis [71]. The presence of BCSCs is also an important cause of ET resistance in luminal breast cancer [72,73], which can be regulated by the cyclin-dependent kinase 4/6 (CDK4/6) complex [74,75] and mechanistic target of rapamycin (mTOR) signaling [76]. The use of CDK4/6 inhibitors (i.e., palbociclib and abemaciclib) or the mTOR inhibitor everolimus significantly improves survival in patients who develop endocrine resistance [77,78,79]. In addition, recent studies have shown that PI3K inhibitors (i.e., alpelisib) have significant clinical activity against PIK3CA-mutated luminal breast cancers, including ET-resistant tumor cells [80]. Combination therapy consisting of BCSC-targeting agents, anticancer agents, and ET may overcome drug resistance and improve therapeutic efficacy; further evaluation of the clinical efficacy and safety of BCSC-targeting agents is needed.

## 8. Host Defense Immunity and Breast Cancer Recurrence

With the exception of the direct cytotoxic/cytostatic effects of anticancer agents and ET on breast cancer cells, host defense immunity against residual tumor cells can be induced by (neo)adjuvant chemotherapy and ET. In the neoadjuvant setting, the activation of immune cells, including NK, CD4+, and CD8+ T cells, at the primary site and increases in the numbers of peripheral immune cells, including pNK and CD8+ T cells, in the blood were observed after NAC, regardless of tumor subtype [81,82]. These findings suggest that the coactivation of local and systemic immune responses plays an important role in the enhancement of the therapeutic effect after NAC. The assembly of peripheral immune cells, such as NK and T cells, in the blood is a key event in the eradication of metastatic breast cancer cells in the axillary lymph nodes in the presence of pre-NAC TILs. How these induced immune responses contribute to the eradication of residual tumor cells, leading to the cure of breast cancer after NAC, is unknown.

Host immune defenses are thought to be modulated by the hypothalamic-pituitary-adrenal (HPA) axis in response to surgical stress and the anesthetic techniques used for breast cancer surgery [83]. General anesthesia (GA) with inhalational anesthetics and opioids suppresses cell-mediated immunity, thereby increasing the recurrence of breast cancer and decreasing the survival rate. The relationship between the anesthetic technique used and cancer recurrence is an interesting and controversial issue in oncological surgery [84,85]. A large prospective RCT revealed no significant difference in the recurrence rate or RFS between patients given GA with sevoflurane/opioids and those given paravertebral block/intravenous (IV) anesthesia with propofol for mastectomy (MX) or breast-conserving surgery (BCS) and axillary lymph-node dissection (ALND) for breast cancer [86]. A subset analysis, however, suggested the existence of a survival effect for patients receiving BCS in Asia, although the follow-up period was inadequate [86]. The HPA axis plays an important role in the regulation of host homeostasis via neuroendocrine mediators such as cortisol, surgical stress, and anesthetics such as inhalational agents; axis activation causes immunosuppression and promotes the regeneration of residual tumor cells, leading to distant metastasis (Figure 4).

Breast cancer surgery has de-escalated from MX and ALND to BCS and sentinel lymph-node biopsy (SLNB), which induce less surgical stress; NAC is applied preoperatively in advanced cases. Currently, most breast cancer cases are managed with BCS and SLNB, followed by adjuvant therapy, depending on the tumor subtype. BCS does not induce significant immunosuppression due to surgical stress, suggesting that the use of IV anesthesia and opioids is involved in the reduction of host defense immunity. Most BCSs can be performed with GA consisting of IV anesthesia with propofol and opioids, such as fentanyl or remifentanil, under mechanical ventilation with tracheal intubation. In contrast, the use of IV anesthesia with low-dose propofol and/or sedation with midazolam without opioids under the maintenance of spontaneous breathing can reduce anesthetic-related immunosuppression. The latter anesthetic technique may improve survival by reducing the recurrence of breast cancer through the maintenance of antitumor immunity after surgical treatment [87]. A prospective RCT examining the effects of BCS performed with and without spontaneous breathing and opioids in patients with breast cancer is needed.

## 9. Conclusions

Despite the development of techniques for the detection of residual tumor cells at the molecular level, the monitoring of these cells in the blood after surgical treatment in clinical practice is not yet possible. The clinical significance of the presence of residual tumor cells and its association with distant recurrence is also currently unknown. The targeting of residual tumor cells based on genomic profiles and the administration of additional adjuvant therapy after NAC comprise one strategy for the reduction of recurrence and improvement of survival in patients with breast cancer. Due to the low frequency of genetic mutations in breast cancer, the analysis of genetic profiles for the design of individual targeted therapies is limited. Instead, elucidation of the gain of antitumor immunity via anticancer agents, the enhancement of drug sensitivity by overcoming drug resistance, and the targeting of therapy based on genomic profiles could lead to the complete curing of primary breast cancer.

## Figures and Tables

**Figure 1 cancers-13-00926-f001:**
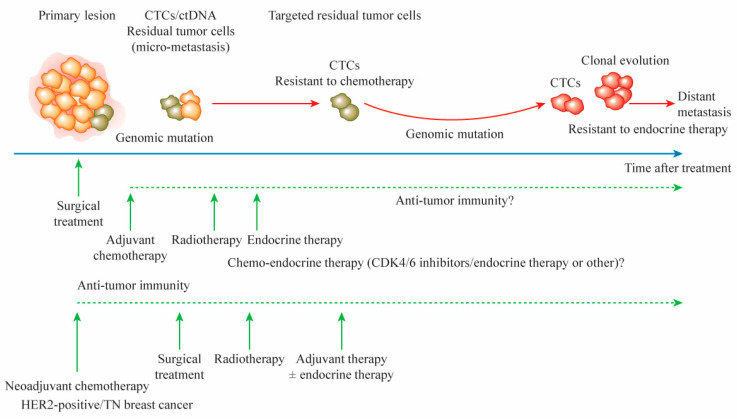
A time course for distant metastasis after adjuvant and neoadjuvant therapies in patients with breast cancer. Although primary lesions and axillary lymph nodes can be removed maximally by surgical treatment, the minimal presence of residual tumor cells, as assessed by CTC and ctDNA analyses, contributes to recurrence and distant metastasis. Genomic mutations in primary tumors can cause resistance to anticancer agents and endocrine therapy. The remaining tumor cells with genomic mutations cause the clonal evolution of tumor growth, leading to distant metastasis. The use of adjuvant therapy in combination with endocrine therapy and CDK4/6 inhibitors may improve survival. After neoadjuvant chemotherapy, additional adjuvant chemotherapy targeting residual tumor cells improves survival. CTC, circulating tumor cell; ctDNA, circulating tumor DNA; CDK, cyclin-dependent kinase; HER-2, human epidermal growth factor receptor 2; TN, triple negative.

**Figure 2 cancers-13-00926-f002:**
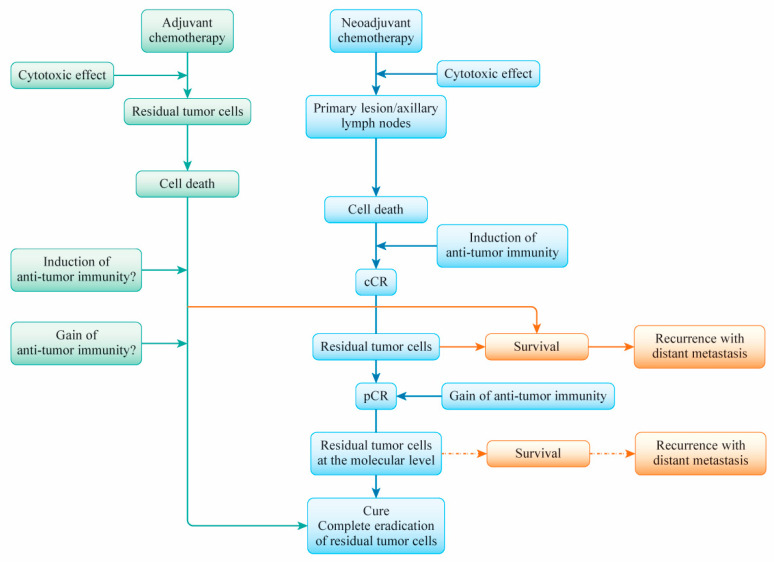
A scheme for the understanding of residual tumor cells’ causing of distant recurrence after neoadjuvant chemotherapy (NAC) or adjuvant chemotherapy in patients with breast cancer. When a patient responds to NAC, the anticancer treatment induces a clinical complete response (cCR) followed by a pathological complete response (pCR). Anticancer agent-induced (immunogenic) cell death evokes the establishment of antitumor immunity, which eventually prevents the regrowth of residual tumor cells. If immunogenic antitumor immunity is not established successfully, meaning that it fails to eradicate residual tumor cells, the presence of these cells at the macro and molecular levels may contribute to distant metastasis. The details of the manner in which antitumor immunity is gained after NAC are not clear. Adjuvant chemotherapy aims to cure breast cancer by targeting residual tumor cells and eradicating them via the cytotoxic effects of anticancer agents. Whether these processes require the induction or gain of antitumor immunity by anticancer agents used in adjuvant chemotherapy is unclear. Drug sensitivity and the establishment of antitumor immunity via anticancer drug therapy may be important for the achievement of a complete cure for primary breast cancer.

**Figure 3 cancers-13-00926-f003:**
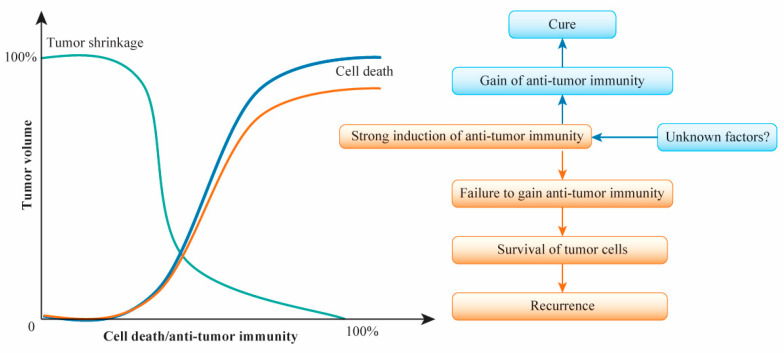
Hypothesized relationships of tumor volume to cell death and antitumor immunity after (neo)adjuvant chemotherapy for primary breast cancer. Decreased tumor volume is hypothesized to be related synergistically to increased cell death and the induction of antitumor immunity. The degree of increased cell death is almost parallel to the degree of antitumor immunity activation. In neoadjuvant chemotherapy responders, tumor shrinkage correlates with the rate of cell death and the induction of antitumor immunity, with the maximum response consisting of clinical complete response leading eventually to pathological complete response. The induction of antitumor immunity leads to the gain of such immunity, with the complete eradication of tumor cells and prevention of distant metastasis. The failure to gain antitumor immunity may allow residual tumor cells to survive, leading to distant metastasis. The gain of antitumor immunity may be regulated by unknown factors.

**Figure 4 cancers-13-00926-f004:**
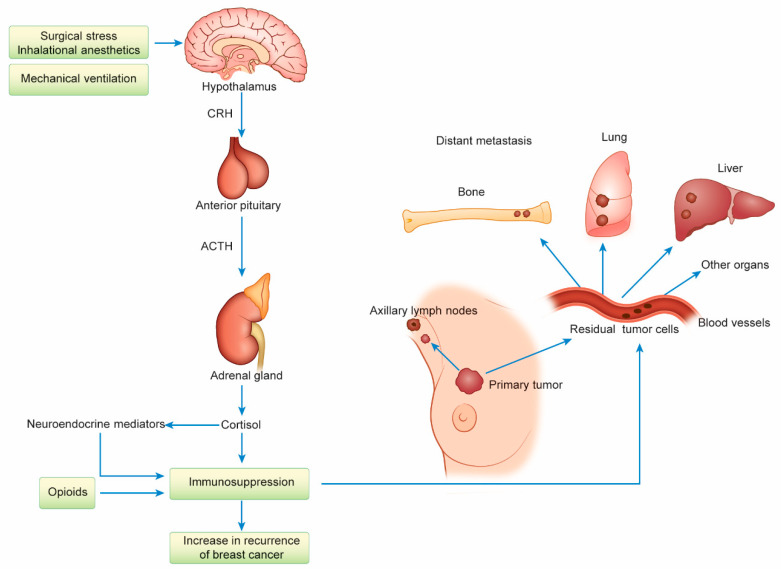
The activation of the hypothalamic-pituitary-adrenal (HPA) axis due to surgical stress and the use of certain anesthetic techniques may be involved in the increased distant recurrence of breast cancer. Surgical stress and general anesthesia (GA) under mechanical ventilation with tracheal intubation activate the HPA axis and increase neuroendocrine mediators, such as cortisol, leading to immunosuppression that increases the distant metastasis of breast cancer. Opioid-induced immunosuppression is also involved in the increase in distant recurrence. The development of distant metastasis to the bone, lung, liver, and other parts of the body from residual tumor cells is observed after surgical treatment. CRH, corticotropin-releasing hormone; ACTH, adrenocorticotropic hormone.

## Data Availability

No new data were created or analyzed in this study. Data sharing is not applicable to this article.

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
