# Peer review of "Clinical Perspectives in Addressing Unsolved Issues in (Neo)Adjuvant Therapy for Primary Breast Cancer"

_cancers, 2021, doi:10.3390/cancers13040926_

Round 1
Reviewer 1 Report
The authors have written a review of the current status of the literature on treatments for early breast cancer, with a particular emphasis on reduction of risk of recurrence due to distant metastasis. The review is well-written, and should be of interest to readers of this journal.
My only concern is the strong statement at line 64: "It goes without saying that the goal of primary breast cancer treatment is the achievement of an absolute cure without recurrence,... ". I think many clinicians would argue that the goal is to improve length and quality of life; this could conceivably be obtained without complete eradication of all cancer cells, as has been shown in men with prostate cancer.
Apart from that, I think the manuscript can be accepted for publication.
Author Response
Thank you for your helpful comments.
We have changed the statement in the text to a generalized and weakened statement (P3, L35-36). In this review, we would like to define cure of primary breast cancer as the eventual complete eradication of cancer cells, regardless of tumor subtype. It is unclear whether the curative status of HR-positive breast cancer can provide a similar situation to that of prostate cancer, with a stable, dormant-like disease with no further growth and longer survival. In metastatic breast cancer, the goal of treatment is not cure, but prolonged survival and improved quality of life, which is different from primary breast cancer.
Reviewer 2 Report
Thank you for your response to my concerns/comments. All my concerns have been addressed.
Author Response
Thank you for your review.
Reviewer 3 Report
The manuscritpt has improved a lot. It was already very well written and organized, although I still found it not so informative due to the subject chosen. In fact, simply there is no answer to most of the problems and questions raised, which limit the possibility to reach a cure in all patients despite new treatments and settings. The lack of a solution is though not resolved and the result is an interesting summary of difficulties to be addressed or that have been partially addressed. For all these reasons the manuscript cannot drow definitive conclusions, but gives insites on how to drive future reasearch in the field. Probably I would change again the title avoiding the sentence"Establishment of a Cure", by saying something like: Clinical Perspectives in Addressing unsolved issues in (Neo)adjuvant Therapy for Primary Breast Cancer.
Author Response
Thank you for your helpful comments.
We have changed the title as you suggested.
Reviewer 4 Report
No further concerns.
Author Response
Thank you for your review.
This manuscript is a resubmission of an earlier submission. The following is a list of the peer review reports and author responses from that submission.
Round 1
Reviewer 1 Report
The manuscript is well-written and easy to follow. All four figures are informative and very helpful. However, there are 51 references in this manuscript and it will be more helpful if the manuscript can provide some descriptive statistics such as proportion, mean and standard deviation of some major variables.
Reviewer 2 Report
Authors highlighted and interpreted a list of unmet needs in breast cancer management and treatment, which is a sort of elegant resume of all the problems limiting the gain of a cure in breast cancer.
Although there is no novelty in the message, nor in the presented data, the presentation is well organized and easily readable.
However, the manuscript is quite superficial and addresses many of the general issues occurring in most of recurring cancers types, without getting into details. Minimal residual disease, primary or acquired resistance, immuno-surveillance and editing are well known major aspects impacting on the cure of cancer: scientists are working on this all over the world.
For all these reasons, the present manuscript appear as an academic presentation of the current updated opinion of the scientific community on questions that still need to be targeted and should be detailed and deepened with some new insigths in order to be interesting for the scientific community.
Reviewer 3 Report
The manuscript is general well written and gives various perspectives on the basic mechanisms and clinical application of neoadjuvant therapy. I have only one major comment.
It would be great if authors provide some current trials along the immune check point therapies. For example, keynote-150 etc. Also, as the authors hint at combinatorial approach, it would be good to cite some trails such as ENHANCE/KEYNOTE, wherein there is combination of chemo with checkpoint.
All-in-all, this is a well written review including some current clinical trails will warrant publication of this manuscript.
Reviewer 4 Report
Authors have given a good narrative review of the current challenges we face in eliminating cancer recurrences. The article does look at various possible mechanism for recurrence and highlights the areas for for future research.
The article will be of interest to the readers. Authors can consider revising the title of the article.
Authors can also consider giving more details about tumour heterogeneity and mechanism of drug resistance against endocrine treatment in breast cancer which can lead to recurrence